# Early Identification of Acute Traumatic Coagulopathy Using Clinical Prediction Tools: A Systematic Review

**DOI:** 10.3390/medicina55100653

**Published:** 2019-09-28

**Authors:** Sophie Thorn, Helge Güting, Marc Maegele, Russell L. Gruen, Biswadev Mitra

**Affiliations:** 1School of Public Health and Preventive Medicine, Monash University, Melbourne 3004, Australia; biswadev.mitra@monash.edu; 2Institute for Research in Operative Medicine, University Witten/Herdecke, 51109 Cologne, Germany; helge.gueting@uni-wh.de (H.G.); marc.maegele@t-online.de (M.M.); 3Department of Traumatology, Orthopaedic Surgery and Sports Traumatology, Cologne-Merheim Medical Centre, 51109 Cologne, Germany; 4ANU Medical School, Australian National University, Canberra 2605, Australia; russell.gruen@anu.edu.au; 5National Trauma Research Institute, Melbourne 3004, Australia; 6Emergency and Trauma Centre, The Alfred Hospital, Melbourne 3004, Australia

**Keywords:** acute traumatic coagulopathy, prediction model, pre-hospital, bleeding, trauma

## Abstract

*Background and objectives:* Prompt identification of patients with acute traumatic coagulopathy (ATC) is necessary to expedite appropriate treatment. An early clinical prediction tool that does not require laboratory testing is a convenient way to estimate risk. Prediction models have been developed, but none are in widespread use. This systematic review aimed to identify and assess accuracy of prediction tools for ATC. *Materials and Methods:* A search of OVID Medline and Embase was performed for articles published between January 1998 and February 2018. We searched for prognostic and predictive studies of coagulopathy in adult trauma patients. Studies that described stand-alone predictive or associated factors were excluded. Studies describing prediction of laboratory-diagnosed ATC were extracted. Performance of these tools was described. *Results*: Six studies were identified describing four different ATC prediction tools. The COAST score uses five prehospital variables (blood pressure, temperature, chest decompression, vehicular entrapment and abdominal injury) and performed with 60% sensitivity and 96% specificity to identify an International Normalised Ratio (INR) of >1.5 on an Australian single centre cohort. TICCS predicted an INR of >1.3 in a small Belgian cohort with 100% sensitivity and 96% specificity based on admissions to resuscitation rooms, blood pressure and injury distribution but performed with an Area under the Receiver Operating Characteristic (AUROC) curve of 0.700 on a German trauma registry validation. Prediction of Acute Coagulopathy of Trauma (PACT) was developed in USA using six weighted variables (shock index, age, mechanism of injury, Glasgow Coma Scale, cardiopulmonary resuscitation, intubation) and predicted an INR of >1.5 with 73.1% sensitivity and 73.8% specificity. The Bayesian network model is an artificial intelligence system that predicted a prothrombin time ratio of >1.2 based on 14 clinical variables with 90% sensitivity and 92% specificity. *Conclusions*: The search for ATC prediction models yielded four scoring systems. While there is some potential to be implemented effectively in clinical practice, none have been sufficiently externally validated to demonstrate associations with patient outcomes. These tools remain useful for research purposes to identify populations at risk of ATC.

## 1. Introduction

Injury is a leading cause of death in the young and middle-aged [1], and uncontrolled haemorrhage is responsible for many of these deaths. Acute traumatic coagulopathy (ATC) is a phenomenon of pathological hypocoagulability post-trauma that is present in about 25% of major trauma patients [2]. The condition has been associated with the excessive activation of the protein C pathway and subsequent reduced activity of coagulation factors and hyperfibrinolysis in addition to decreased clot formation [3]. The development of ATC is correlated strongly with the presence of blunt tissue trauma and hypoperfusion [4].

ATC has been associated with high mortality, particularly when accompanied by hypothermia and acidosis; a difficult to treat state known as the “triad of death” [5]. Currently, diagnosis of ATC is by conventional laboratory coagulation tests, which anecdotally take about 45 min to return, or viscoelastic measures of coagulation, which are not available in all centres. During this time, worsening coagulopathy may contribute to ongoing shock, leading to a vicious cycle and therefore, specific treatment should ideally be started towards prevention and early management of ATC [6]. In contrast to formulaic empirical management of all patients, which may be inefficient and associated with adverse effects, a clinical prediction tool that can identify coagulopathic patients on or before arrival at hospital would be valuable to facilitate such early management.

The aim of this systematic review was to identify published ATC prediction models, evaluate their accuracy and discuss limitations and strengths of each model. We assessed the feasibility of implementing each tool and limitations in study design and model development.

## 2. Materials and Methods

This systematic review was performed in accordance with the 2015 Preferred Reporting Items for Systematic Reviews and Meta-Analyses (PRISMA) statement [7] (see Appendix A). The study protocol was registered with the International Prospective Register of Systematic Reviews (Prospero) database (registration number: 42018090731). As only published material was used to construct this review, no ethics committee approval was required. Tables describing the included scoring systems and tools are reproduced from their original publications and cited as such.

### 2.1. Search Strategy

We performed our literature search on February 23rd, 2018. We searched OVID Medline (1946 to Present with Daily Update) and Embase (Embase Classic + Embase 1947 to search date) using a Boolean search strategy. We used Medical Subject Headings (MeSH) terms and keyword searches relating to two broad topics: The population, trauma patients, and the outcome, coagulopathy. We applied a filter to the search to restrict results to predictive, prognostic and observational studies. We combined synonyms with “OR” and concepts with “AND”, and limited the search to studies published in English, as it was beyond the scope of the researchers to translate other languages. The full search strategies for Medline and Embase are available in Appendix B and Appendix C. Grey literature was identified by checking references of eligible studies: Two authors read the reference lists of included articles and screened any potentially relevant articles which had not yet been found for inclusion. Two relevant records were available only as abstracts from conference publications. We contacted the lead authors directly and were supplied with a poster [8] and PhD thesis [9], respectively, for these studies.

### 2.2. Study Selection

Identified records were scrutinised by two independent reviewers (ST and HG) and evaluated based on pre-discussed inclusion and exclusion criteria during abstract or full article screening. Any discrepancies between the two reviewers were decided by a senior reviewer (MM). We chose to define ATC based on laboratory diagnosis rather than requirement of massive transfusion, as the population of patients with ATC and those requiring massive transfusion are not identical [10]. Risk of bias and concerns about applicability were assessed using the Prediction model study Risk Of Bias ASsessment Tool (PROBAST) [11].

### 2.3. Inclusion & Exclusion Criteria

Articles were included if they explained the development and/or validation of a scoring system comprised of more than one clinical variable that estimated the likelihood of any trauma patient developing ATC. Single variable studies were not included as they did not reflect scoring systems. Studies pertaining to primarily surgical, cardiac or septic patients were excluded. The outcome of interest was laboratory-based diagnosis of ATC, i.e., coagulopathy defined by International Normalised Ratio (INR), prothrombin time ratio (PTr), activated partial thromboplastin time (aPTT), thromboelastography (TEG) or rotational thromboelastometry (ROTEM) or fibrinogen levels. We included studies with patients 16 years and older.

## 3. Results

The initial search strategy identified 3096 records after the removal of duplicates. We identified an additional 9 records during text-mining. Through title and abstract screening, we excluded 3020 records. We included 85 articles in full to assess suitability, of which 67 articles were excluded in full text screening for including paediatrics, being only available in abstract or referring to a different outcome. Eighteen articles discussed predictive and associated factors of ATC using the outcome of laboratory-based coagulopathy. Among these, three full text articles described ATC prediction tools, as well as two conference abstracts. One abstract [8] was accompanied by a poster publication which describes an early version of a score also available in full text [12], and the other introduced a score that is discussed in a PhD thesis [9,13]. One additional article described the validation of a scoring system previously produced to predict ATC. Inclusion of the six studies for this review is illustrated in the PRISMA Flow Diagram (Figure 1). Table 1 demonstrates the results of applying the PROBAST to the included studies [14]. A meta-analysis was deemed inappropriate due to the substantial variation in cohorts, outcome definitions and study aims between reported studies.

### 3.1. Records Identified

Each of the included studies was evaluated based on the Transparent Reporting of a multivariable prediction model for Individual Prognosis or Diagnosis (TRIPOD) Statement [15] and discussed below. The six studies included in the present review are summarised in Table 2.

### 3.2. The COAgulopathy of Severe Trauma (COAST) Score 

The COAST score, the first prehospital ATC clinical prediction tool to be published, was retrospectively developed on 1680 Australian patients in 2006–08 and prospectively validated in the same centre on 1225 patients in 2009 (Table 3) [16]. Major trauma patients with Injury Severity Score (ISS) >15 requiring urgent surgery, ICU admission, or who died were included and secondary admission, prehospital blood products, or operative management and incomplete data were exclusion criteria. ATC was defined as INR > 1.5 or aPTT > 60 s and the prevalence of ATC was 9.0% and 8.2% in the development and validation cohorts, respectively.

### 3.3. SBP: Systolic Blood Pressure

A score ≥3 points was defined as a positive COAST score, as this represented the best balance between sensitivity and specificity. The COAST score performed on internal validation with 60% sensitivity, 96.4% specificity, 93.3% accuracy and an AUROC of 0.83 (95% CI 0.78–0.88).

#### 3.3.1. Limitations

The tool was developed and validated in the same single centre in Victoria. No external validation has been undertaken. The selection of the predictor variables was not explained. Head injury due to motor vehicle crashes was very common in this population, which may have contributed to head injury being non-predictive of coagulopathy. There were variables associated with coagulopathy that were not included in the score: Prehospital cardiopulmonary resuscitation had an odds ratio of 13.2, (*p* < 0.001); however, only 20 patients received CPR. Prehospital intubation also conferred increased risk (OR 7.3, *p* < 0.001), with 154 intubated patients. Chest injury was predictive (OR 1.95, *p* = 0.001) but was not included because it is difficult to identify and grade the severity of chest injuries in the prehospital setting.

In the derivation cohort, average temperature was 33.4 ± 9.8 °C whereas in the validation cohort it was 36.3 ± 1.2 °C. The large discrepancy and unusually wide standard deviation suggest that missing data may have been incorrectly recorded as 0 or −1 °C. This wide range was not replicated in the prospectively collected validation cohort. In the validation cohort, missing data was dealt with by “listwise deletion of cases”, which was not done in the derivation cohort.

Abdominal or pelvic content injury had to be diagnosed by paramedics without diagnostic machinery. Although the prehospital paramedics were quite accurate [16], a subjective measure is not without problems. A threshold for injury severity, e.g., using the Abbreviated Injury Scale, was not recorded in the score description.

#### 3.3.2. Strengths

The score is simple and can be easily memorised and calculated without technological tools or diagnostic equipment in the prehospital phase, during triage, or soon after arrival in the emergency department (ED). The COAST score had a high specificity at cut-off ≥3 (96.4%), which results in little over-diagnosis. In the validation cohort, correctly identified patients with coagulopathy had significantly worse outcomes, including more urgent surgery (55.0% vs. 25.0%) and higher mortality (53.3% vs. 27.5%) than missed patients, which demonstrated the predictive ability of the COAST score for severe cases.

### 3.4. Trauma-Induced Coagulopathy Clinical Score (TICCS)

TICCS was developed on a prospective single-centre Belgian cohort of 82 patients in 2012–13 and validated retrospectively on 33,385 cases from the German Trauma Registry (TR-DGU) [17,18]. Included cases were aged ≥16 years with an ISS of >8 and admission to a resuscitation room, excluding secondary admissions or cases without required data. The development patient outcome was either “severe” (TEG/ROTEM-diagnosed coagulopathy OR INR >1.3 OR fibrinogen <1.5 g/L AND all three of persistent hypotension with active bleeding, massive transfusion and surgical or endovascular haemostasis procedure) or “non-severe” (not fulfilling all severe criteria). The score was developed based on the evidence that tissue injury and hypoperfusion are the primary drivers of ATC [3,19]. The score was modified for validation on the TR-DGU and the outcome of interest on this validation was blood transfusion. The original and modified scores are shown in Table 4.

In the development cohort, eight (9.8%) patients were severe. With a cut-off of ≥10, TICCS correctly identified all eight severe cases, but had three false positives. AUROC was 0.98, sensitivity was 100%, specificity 95.9%, positive predictive value (PPV) 71.7% and negative predictive value (NPV) 100%. All of the false positives had laboratory-defined coagulopathy but did not fulfil other “severe” criteria.

In the validation cohort, 12.7% of patients required blood transfusions. With a cut-off of ≥12, TICCS had an AUROC of 0.700 with a PPV of 48.4%, NPV 89.1%, sensitivity 17.7% and specificity 97.3%.

#### 3.4.1. Limitations

The development and validation studies for TICCS had many inconsistencies, including population characteristics, prediction tool variables and measured outcome. In the development study, the inclusion and exclusion criteria were not fully explained. Furthermore, the sample was small and from a single hospital. The population only included blunt trauma patients. The outcome of interest was chosen for its relationship to massive transfusion, not ATC.

The validation study, which was retrospective and therefore more difficult to fit the prediction tool to, was large enough to demonstrate efficacy, but the score demonstrated a lower PPV. In the validation study, only patients with an ISS of >8 were included, unlike in the original study, which had an ISS range of 4–66. Significant modifications were required for the validation, which calls into question the utility of the validation.

#### 3.4.2. Strengths

TICCS is a simple score, can be calculated by paramedics before arrival at the hospital and gives an idea of overall injury severity on arrival. The development of the scoring variables was sound; using knowledge that tissue injury and hypoperfusion are the main drivers of ATC reduces confounding and the impact of chance, particularly in a small development cohort. The validation cohort was large enough that any outcomes were statistically significant, and the retrospective study design minimises selection or detection bias. The high NPV demonstrates the utility of TICCS for ruling out patients’ need for blood transfusions; however, its low PPV indicates the need for combination with another score or test before initiating treatment.

### 3.5. Prediction of Acute Coagulopathy of Trauma (PACT) 

The PACT score was developed on a single-centre prospective cohort of 315 patients but underwent significant changes before a second multi-centre retrospective development study on 1963 patients and subsequent single-centre prospective validation on 285 patients [12]. There were differences in inclusion and exclusion criteria between development and validation cohorts, including packed red blood cells (PRBC) requirement (inclusion), blunt trauma (inclusion), isolated hip fracture (exclusion), prehospital blood transfusion (exclusion) and expected survival <24 h (exclusion). The outcome of interest was an INR of >1.5. The PACT score is shown in Table 5 [12].

At a threshold of ≥196 points for ATC, the PACT score had a sensitivity of 73.1% and specificity of 73.8%. The overall AUROC of the score was 0.74 on the derivation cohort and 0.80 on validation.

#### 3.5.1. Limitations

There were several differences between the derivation and validation populations: The average ISS was 17.3 (standard deviation (SD) 12.0) in derivation and 32.3 (SD 15.1) in validation, a greater proportion of the validation cohort were intubated (50.9% vs. 16.2%) and more patients were coagulopathic (validation = 9.1%, development = 5.9%). This may be as a result of different inclusion criteria or changing guidelines.

In the validation cohort, only patients who had received a blood transfusion were included. In the development cohort, penetrating injuries were not excluded, although the USA has approximately twice as many penetrating injuries as European countries do (7.9% vs. 4.2%) [20,21], and indeed 12% of the development cohort had penetrating trauma.

#### 3.5.2. Strengths

The PACT score is simple and can be calculated pre-hospital, albeit probably not possible for it to be memorised and calculated without an aid. The sample size was maintained in the validation cohort by imputing missing data or using surrogate markers. The score was validated on an independent population and the technicians investigating the INR were blinded to other patient data.

Exclusion criteria in the validation study were thorough; notably, anticoagulated patients and patients with isolated traumatic brain injury (TBI) were excluded, which targeted the outcome to conventional ATC only, not TBI-induced.

### 3.6. Bayesian Network Model 

Perkins et al. developed the Bayesian network model, a predictive system that “learns” the value of variables through artificial intelligence technology, on 600 cases from the Activation of Coagulation and Inflammation in Trauma (ACIT) study, a multi-national prospective cohort study [9]. It was then prospectively validated on 491 cases from three hospitals. Included patients were >15 years old, required trauma team activation and were primary admissions. Cases were excluded if injury was >2 h before arrival, they received >2000mL of prehospital IV fluid, had burns covering >5% total body surface area, were on anticoagulants, declined consent or had liver disease or a bleeding diathesis. ATC was defined as a PTr of >1.2 plus algorithm-recognition and expert opinion. The Bayesian Network Model is described in Table 6.

With sensitivity set at 90%, the model had 82% and 92% specificity on development and validation cohorts, respectively, and an AUROC of 0.927 and 0.964 respectively. Furthermore, a significant difference was evident in outcomes between coagulopathic and non-coagulopathic patients, as shown in Table 7.

The model was also assessed for performance when variables were missing, as is often the case in emergency patients, and the accuracy was not significantly affected. Variables representing hypoperfusion (blood pressure, heart rate, lactate, base deficit and pH) were the most accurate predictors, although the multivariate score was more accurate than any variable alone.

#### 3.6.1. Limitations

This is a strong, well-designed study with few major limitations. Advanced technology was used to develop the tool and the system relies on this technology to function. The Bayesian Network score is calculated using an online calculator. It would not always be possible to calculate this score in the pre-hospital phase because of its reliance on blood gas analysis, however it was included in this review because blood gases are sometimes available before arrival at hospital, and if not, they are likely to be performed immediately upon arrival.

The system “learnt” from the characteristics of the input population, the ACIT study [22]. The population characteristics, while not necessarily universal, are unlikely to be differ substantially from major trauma populations elsewhere in the developed world.

Some of the data points may be unreliable: Temperature was missing in about 40% of cases; and patients who received >2000 mL of IV fluid pre-hospital were excluded, so information pertaining to this population was imputed.

In order to attain a sensitivity of >90%, a lower specificity was accepted for this tool. 80% of the false-positives in the two cohorts were patients with catastrophic head injuries (abbreviated injury scale (AIS) ≥ 5), who had universally bad outcomes regardless of coagulation status.

#### 3.6.2. Strengths

The Bayesian Network model is highly accurate. It is not significantly affected by missing data and has little risk of over-fitting because the predictors were based on prior knowledge rather than database analysis. This is a particular strength because some of the variables may not be available on hospital arrival.

The outcome of interest in this study was specific and thorough—conventional laboratory diagnosis of coagulopathy identified 94.2% of coagulopathic patients, but the remaining 5.8% had normal PTr with other evidence of coagulopathy identified by physicians.

## 4. Discussion

The studies on prediction tools for ATC were appropriately developed and reported; each article fulfilled most of the criteria on the TRIPOD Statement [15] for good reporting of prognostic models. Adequate statistical analysis and performance testing was undertaken, and limitations to the models were freely discussed by the authors in their publications. Predictors were developed through a combination of clinical and statistical reasoning, although their selection was not always adequately explained.

None of the prediction tools are ready for widespread clinical use, as no model has performed well enough on external validation to justify including it in clinical guidelines. We attempted to assess for publication bias by searching clinicaltrials.gov for unpublished studies but we found no other trials on ATC prediction scores. Prediction models often perform worse on external validation than during development or internal validation [23], so the tools may need slight adjustments for different populations. Excessive altering can invalidate the system: For example, TICCS performed poorly when validated on the TR-DGU possibly because the variables required too much alteration. It was arguably a different score that was being validated. The COAST, TICCS and PACT scores were all suitable to be applied to patients before arrival at the hospital, as they do not require the results of blood tests to be calculated. The Bayesian model, while highly sensitive and specific, requires information that is not available to pre-hospital personnel. This calls into question its utility as an “early” identification tool; although it still provides useful information very early during hospitalisation, it cannot guide pre-hospital administration of tranexamic acid, a treatment for ATC currently under investigation. We included the Bayesian model in our review because of its potential utility in expediting administration of treatment upon arrival in the hospital.

Scores that are developed on small or single-centre cohorts may not generalise well to other populations. TICCS was developed on a small single centre and performed poorly on the large validation cohort. PACT had very different development and validation cohort characteristics but had a multi-centric development. COAST was developed and validated on the same single centre so it requires external validation before being deemed generalizable.

Among the included studies, the decision of whether or not to exclude cases with known, pre-existing coagulation disorders, such as chronic liver disease or anticoagulant therapy, was inconsistent. The PACT score and the Bayesian model excluded these groups, which may produce a purer sample of patients with only “true” ATC, as opposed to injured people with coagulopathy of other aetiology. However, as this information may not be readily available in the pre-hospital or early in-hospital environment, the scores which are adapted to suit any patient presenting after injury, regardless of these factors, may be more useful. Additionally, developing these scores retrospectively eliminates attrition bias and is cheaper and easier to execute than prospective studies.

The issue of how to deal with missing data is significant in all studies as it introduces considerable selection bias to exclude all patients with a missing variable. This is particularly relevant for scores developed retrospectively using registry data [24]. The Bayesian network showed that its accuracy was not affected by missing data points but COAST, TICCS and PACT relied heavily on few variables. Multiple imputation was used in COAST, TICCS and PACT to substitute missing data with imputed data, and this has been previously demonstrated to reduce selection bias [25].

The definition of ATC varied among these studies, and indeed no universal definition exists. However, it has been previously demonstrated that, regardless of the thresholds used, designating a defined threshold for ATC would be beneficial when designing studies on ATC and also when comparing studies in a systematic review. Attempts have been made to do this, but thus far the differing goals and perceptions of ATC (i.e., whether any coagulopathy is significant, or only coagulopathy in combination with severe haemorrhage-related outcomes) have prevented a resolution. An INR of >1.5 or an aPTT of >60 s is a comparatively high threshold for diagnosis of ATC. ATC is often defined by a PTr/INR of >1.2 [13,26,27] or an INR of >1.3, [17,28,29,30,31] and less commonly by an INR of >1.5. [12,16] An APTT-based definition of ATC also varies widely, from >35 s [26] to >60 s [16,32,33]. In a study aiming to identify the most useful threshold [34], 1031 trauma patients who required PRBC transfusion had their INR tested—50% of patients had an INR of >1.2, for whom in-hospital mortality was 18.5% and massive transfusion rate 32.2%; whereas 21% of patients had an INR of >1.5 and they had an in-hospital mortality of 26.5% and massive transfusion rate of 45.2%. This shows that the population with an INR of >1.5 have significantly worse outcomes, however the clinical significance of coagulopathy at an INR of 1.2–1.5 remains unknown. This study is also confounded by only including patients who required PRBCs. The outcome heterogeneity between the studies suggests that a diagnosis of ATC based solely on an INR/aPTT is incomplete and that the outcome should include clinical manifestations of haemorrhagic progression [35]. Many prediction scores exist to predict the requirement of massive transfusion as a pragmatic, management-focussed outcome. The aim of our systematic review was to predict the coagulopathy itself so we chose to use primarily the INR/aPTT as outcomes. Future prediction scores and systematic reviews should consider using a multifactorial outcome which better identifies significant coagulopathy, such as in the Bayesian Network system [9].

Thromboelastography is becoming more common and useful in the diagnosis of ATC but was not used as an outcome because it is not yet ubiquitous in trauma centres. Point-of-care INR tests are also becoming more common but do not correlate sufficiently well with conventional INR tests to be relied upon [36]. Thromboelastography and point-of-care tests may in the future obviate the need for ATC clinical prediction tools, but prediction tools can be cheap, quick and useful. Additionally, measuring blood transfusion rates as the outcome is not specific to ATC, as it does not discriminate between patients who are bleeding but not coagulopathic and those who are coagulopathic. Using a prediction tool calibrated to this outcome would result in over-diagnosis. The Bayesian model combines laboratory tests and clinical findings to avoid missing cases [37]. A combination outcome of clinical and biochemical evidence of coagulopathy is preferable, although more complicated.

The balance between sensitivity and specificity is an issue in developing prediction tools. In COAST, the aim was to identify a group with worse outcomes who may benefit from directed treatment for ATC, therefore high specificity was prioritised. Conversely, the Bayesian model aimed to avoid missing cases so prioritised high sensitivity. This “over-triage” [9] may be harmful as the treatment of ATC using blood products can be expensive or have deleterious effects on people with normal coagulation [10]. However, lower sensitivity can result in under-treatment.

A factor that was very much considered in the development of these tools was the simplicity of its use; if a score is too difficult or time-consuming to calculate in the emergency department, it is not feasible. The COAST score and TICCS are simple enough to be calculated by hand using a short checklist before or upon hospital arrival. PACT is somewhat more complicated but can also be calculated manually. The Bayesian network uses initial ED blood tests and requires the input of variables into an online calculator, making it more complex than other scores.

This systematic review is limited by wide variation between the tools discussed. The cases had different injury severities, inclusion and exclusion criteria, outcomes and goals. Injuries to certain body regions particularly increase risk of coagulopathy; therefore, if more studies were available it would be of interest to stratify by body region. For example, TBI is thought to cause a different form of coagulopathy [38]. TICCS was designed to identify the “severe” patient, as described above, while the other scores looked for just coagulopathy in the post-trauma period. Ideally, we would be able to compare more similar tools; however, with so few ATC prediction tools in publication, we think this review is nevertheless useful.

The high mortality and persistent difficulties in treating ATC make a reliable clinical prediction tool essential. TEG/ROTEM technology is advancing quickly and may soon be the gold standard for diagnosing ATC. Our current reliance on slow laboratory tests is suboptimal. The four clinical prediction tools discussed here, as well as many prediction models for massive transfusion, have been developed around the world, with relatively few attempts to validate the scores on different populations. External validation is vital to minimise selection bias, ensure validity and to improve international consistency in the quality of treatment trauma patients receive. However, prediction scores are unlikely to perform equally well on all populations, particularly when the mechanism of injuries and management protocols vary between countries. Moving forward with the widespread introduction of clinical prediction scores, the most effective route may be for different centres to validate different scores and find one which suits their population. Continuing to develop new scores, which are likely to include similar variables, is an inefficient use of resources. Common themes among scores are markers of hypoperfusion (hypotension, base deficit) and evidence of severe tissue trauma (high ISS, internal bleeding). Using Bayesian technology to develop models may produce the most accurate scores, but they are not the easiest to use and may not yet be appropriate for use.

The purpose of this review was to identify and bring to light existing ATC prediction models, to encourage the validation, implementation and use of effective models and the abandonment of inadequate scores.

## 5. Conclusions

The ATC prediction models discussed are based on solid evidence and have sound study designs. They have differing levels of accuracy; however, none have been sufficiently validated to be deemed reliable and widely applicable. Validating these tools across multiple large trauma centres, using a common definition of ATC, is the next step in facilitating earlier effective treatment of coagulopathy and reducing trauma mortality.

## Figures and Tables

**Figure 1 medicina-55-00653-f001:**
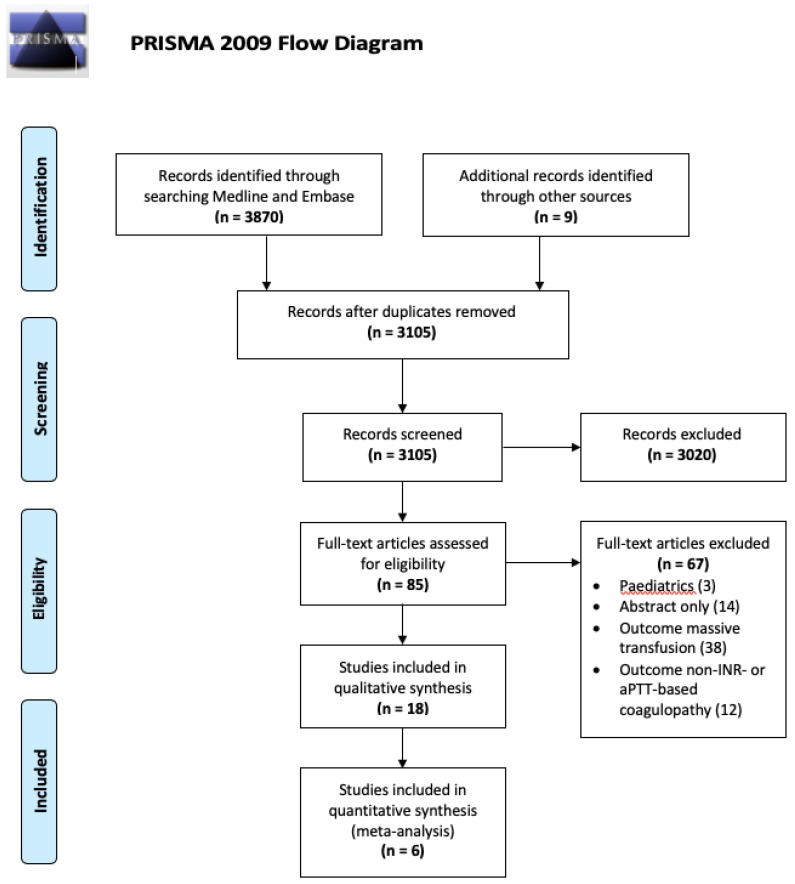
PRISMA flow diagram demonstrating the inclusion and exclusion process undertaken.

**Table 1 medicina-55-00653-t001:** PROBAST results.

Study	Risk of Bias				Applicability			Overall	
	Participants	Predictors	Outcome	Analysis	Participants	Predictors	Outcome	ROB	Applicability
COAST	+	+	+	+	+	+	+	+	+
TICCS 1	−	+	+	−	+	+	?	−	?
TICCS 2	+	+	+	+	+	+	+	+	+
PACT 1	+	+	+	?	+	+	+	?	+
PACT 2	+	+	+	+	+	+	+	+	+
Perkins	+	+	+	+	+	+	+	+	+

PROBAST—Prediction model Risk of Bias Assessment Tool; ROB—risk of bias. + indicates low ROB/low concern regarding applicability; − indicates high ROB/high concern regarding applicability; ? indicates unclear ROB/unclear concern regarding applicability.

**Table 2 medicina-55-00653-t002:** Relevant records identified on systematic literature search.

Name (year)	Score	Purpose	Score Variables	Development/Validation Patient No.	Cohort	Outcome	Sensitivity Specificity AUROC
Mitra (2011) [16]	COAST (COAgulopathy of Severe Trauma)	Retrospective cohort analysis (development) and prospective cohort study (validation)	Entrapment Prehospital SBP Temperature Chest decompression Abdominal or pelvic injury	1680/1225	Single level 1 trauma centre Victoria, Australia	INR >1.5 or aPTT >60 s	60.0% 96.4% 0.83
Perkins (2013) [9] PhD thesis	Bayesian network model	Retrospective cohort analysis (development) and prospective and retrospective cohort analyses (validation)	HR Temperature SBP Haemothorax FAST Pelvic or long bone fracture GCS Lactate Base deficit pH Injury mechanism and energy IV fluid volume	600/373	*Activation of Coagulation and Inflammation in Trauma* study cohort in London, UK (development) and a trauma centre in Oxford, UK, and Cologne, Germany (validation)	PTr >1.2	90.0% 92.0% 0.96
Tonglet (2014) [17]	TICCS (Trauma-Induced Coagulopathy Clinical Score)	Prospective non-controlled observational clinical study	Resuscitation room admission SBP Distribution of injuries	82/–	Single level 1 trauma centre in Liège, Belgium	INR >1.3 or fibrinogen <1.5 or ROTEM-defined coagulopathy	100% 95.9% 0.98
Peltan (2015) [8] Poster	PACT (Prediction of Acute Traumatic Coagulopathy)	Prospective cohort study	Shock index Injury-to-ED time White race Age First GCS First RR	324/–	Single level 1 trauma centre in Washington, USA	INR ≥1.5	73% 74% 0.79
Peltan (2016) [12]	PACT	Retrospective cohort analysis (development) and prospective cohort study (validation)	Shock index Age GCS Mechanism of injury Intubation CPR	1963/285	44 trauma centres on the Oregon Trauma Registry	INR >1.5	73.1% 73.8% 0.74
Tonglet (2017) [18]	TICCS	Retrospective cohort analysis	Admission to resuscitation room SBP Distribution of injuries	–/33,385	Over 600 trauma centres on the German Trauma Registry	Blood transfusion	48.4% 89.1% 0.83

Important information on each publication is summarised here. AUROC—Area under the Receiver Operating Characteristic Curve. SBP—systolic blood pressure; INR—International Normalised Ratio; aPTT—activated Partial Thromboplastin Time; HR—heart rate; FAST—Focused Assessment with Sonography in Trauma; GCS—Glasgow Coma Scale; PTr—prothrombin ratio; ROTEM—rotational thromboelastometry; ED—emergency department; RR—respiratory rate; CPR—cardiopulmonary resuscitation.

**Table 3 medicina-55-00653-t003:** The COAgulopathy of Severe Trauma (COAST) score.

Variable	Value	Score
Entrapment	Yes	1
Systolic blood pressure (SBP)	<100 mmHg <90 mmHg	1 2
Temperature	<35 °C <32 °C	1 2
Chest decompression	Yes	1
Abdominal or pelvic content injury	Yes	1
Highest total possible		7

**Table 4 medicina-55-00653-t004:** The original and modified Trauma-Induced Coagulopathy Clinical Score (TICCS) score.

Original Criteria	Points	Modified Criteria	Points
General severity Admitted to resus room Regular ED room	2 0	General severity Admitted to resus room	2
Blood pressure <90 at least once Always >90mmHg	5 0	Blood pressure <90 at least once Always >90mmHg	5 0
Significant injuries Head/neck L upper extremity R upper extremity L lower extremity R lower extremity Torso Abdomen Pelvis	1 1 1 1 1 2 2 2	Significant injuries (AIS ≥3) Head/neck Upper extremity Lower extremity Torso Abdomen Pelvis	1 1 1 2 2 2
Total possible score	0–18	Total possible score	2–16

ED—emergency department; resus room—resuscitation room; AIS—Abbreviated Injury Scale; R—right; L—left.

**Table 5 medicina-55-00653-t005:** The PACT Score.

Variable	Value	Points Per Unit
Prehospital shock index ≥1	Yes/no	90
Age	Years to nearest decade	1
Mechanism of injury not motor vehicle, motorcycle or bicycle crash	Yes/no	50
Number of GCS points below 15	15-GCS	7
Prehospital CPR	Yes/no	120
Prehospital intubation	Yes/no	50
Total possible points	-	394 + age

Shock index—heart rate/systolic blood pressure; GCS—Glasgow Coma Scale; CPR—cardiopulmonary resuscitation.

**Table 6 medicina-55-00653-t006:** The Bayesian Network Model.

Variable	Value
Heart rate	Beats per minute
Systolic blood pressure	mmHg
Temperature	≥34 °C or <34 °C
Haemothorax	Present (suspected, or on CXR) or absent
FAST	Positive for abdominal free fluid or negative
Unstable pelvic fracture	Present (suspected, or on PXR) or absent
Long bone (femur, tibia, humerus) fracture	Present (suspected) or absent
GCS	On admission or prior to intubation
Lactate	On arterial or venous blood gas analysis
Base deficit	On arterial or venous blood gas analysis
pH	On arterial or venous blood gas analysis
Mechanism of injury	Blunt or penetrating
Energy	High (gun-shot wound, fall >6 m, vehicular mechanism, entrapment, crush or blast) or low (stab, other blunt injury, low-velocity gun-shot)
Volume of fluid administered	<500 mL or 500–2000 mL or >2000 mL crystalloid or colloid

CXR—chest x-ray; FAST—focused assessment with sonography in trauma; PXR—pelvic x-ray; GCS—Glasgow Coma Scale.

**Table 7 medicina-55-00653-t007:** Outcomes in the development and validation cohorts analysed for the Bayesian Network Model.

	Development		Validation	
Outcome	Coagulopathy	No Coagulopathy	Coagulopathy	No Coagulopathy
24-hour mortality	36.6%	1.3%	24.5%	0.2%
In-hospital mortality	53.7%	5.6%	49.1%	5.2%
Blood transfusion	90.1%	23.2%	96.2%	16.0%
Massive transfusion*	40.9%	1.3%	47.2%	0.9%

* Massive transfusion is defined as ≥10 units of packed red blood cells within 24 hours.

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
