# Peer review of "Early Identification of Acute Traumatic Coagulopathy Using Clinical Prediction Tools: A Systematic Review"

_medicina, 2019, doi:10.3390/medicina55100653_

Round 1
Reviewer 1 Report
Thank you for the opportunity to review this manuscript. Overall I found it interesting, but I have some concerns that need to be addressed before it is ready for publication:
Abstract
I think you need to write out some of the words in full (e.g Glasgow Coma Scale not GCS). Also write CPR and PTr in full.
In the conclusion of the abstract, “The search ATC prediction models yielded four prediction tools” does not make sense. Please re-phrase.
Introduction
Line 39 “in young and middle age..”. The authors either mean “in the young and middle aged” or “in young and middle aged patients”.
Line 44 “ATC appears to be present only in patients with a combination of blunt tissue trauma and hypoperfusion”. This is not true, and is not supported by reference 4!
Lines 46-47: ATC is part of the triad of death, rather than complicated by it.
Line 48: “45 minutes to return”: do the authors have a reference for this? Or is it their anecdotal experience?
Line 49: “which remain unavailable in most centres”. Again, do the authors have a reference for this bold statement? I’m not sure that it is even true. And in which country? Do they mean internationally?
Line 52: “blind formulaic management” sounds a bit pejorative. Do the authors mean empirical treatment?
Methods
Lines 80-81: “the population of patients with ATC and those requiring massive transfusion are substantially different.” This is not true, and not supported by reference 10.
In the inclusion criteria the authors say that “coagulopathy defined by International Normalised Ratio (INR), activated partial thromboplastin time (aPTT), thromboelastography (TEG), or rotational thromboelastometry (ROTEM) or fibrinogen levels.” But then later the authors include the Zane Perkins PhD thesis which uses PTr. The methods and results needs to match up better.
Results
I don’t understand how the authors have excluded some studies that were abstracts only (eligibility phase of the PRISMA diagram), but then in their final 6 studies they ended up with two abstracts (one had an associated poster and one had a PhD thesis). This doesn’t make sense. Did they contact all of the authors of the abstracts to see if there was an associated poster or PhD, or only those two? If so, why? Did the previous abstracts lack eligibility in other ways?
Does the second Tonglet study (2017) belong with the main studies of interest? It’s outcome is based on blood transfusion rather than coagulopathy. Therefore it is rightly included in this manuscript in terms of the validation of the earlier score, but is does not itself fulfil all the eligibility criteria. I would therefore say that the authors have identified 5 (not 6) studies that fulfilled all eligibility criteria.
For the tables that reproduce the various scoring systems (Table 3 onwards), have the authors taken these directly from the studies of interest (in which case they should declare it), or have they adapted the tables, or made their own based on the individual studies? This needs to be more clear and avoid any accidental plagiarism
Line 192 “man” is a typo.
Line 306: “The PACT score and the Bayesian model excluded these groups, which may produce a purer sample of patients with only “true” ATC”. I don’t understand what the authors mean here – they need to elaborate or delete this statement.
Line 324 “INR >1.5 or aPTT >60sec is a comparatively high threshold for diagnosis of ATC”: do the authors have a reference to support this statement? Have they just taken the upper ends of the values quoted in the subsequent statements? This seems a bit arbitrary.
Line 344 I don’t think the authors can declare that TEG and point of care tests are “ineffective and relatively uncommon” without some elaboration or citing some sources for this bold statement! There are many centres (for example in the USA) that use TEG and not ROTEM.
Discussion
Author Response
Thank you for the opportunity to review this manuscript. Overall I found it interesting, but I have some concerns that need to be addressed before it is ready for publication:
Abstract
I think you need to write out some of the words in full (e.g Glasgow Coma Scale not GCS). Also write CPR and PTr in full.
We thank the reviewer for this suggestion, full terms have been written before abbreviations.
In the conclusion of the abstract, “The search ATC prediction models yielded four prediction tools” does not make sense. Please re-phrase.
Thank you for noticing this error, the sentence has been corrected to “The search for ATC prediction models yielded four scoring systems.”
Introduction
Line 39 “in young and middle age..”. The authors either mean “in the young and middle aged” or “in young and middle aged patients”.
We appreciate this note, the phrase has been altered to “in the young and middle-aged”.
Line 44 “ATC appears to be present only in patients with a combination of blunt tissue trauma and hypoperfusion”. This is not true, and is not supported by reference 4!
We thank the reviewer for noting this discrepancy. The sentence has been rewritten to more closely reflect the findings of reference 4: “The development of ATC is correlated strongly with the presence of blunt tissue trauma and hypoperfusion”
Lines 46-47: ATC is part of the triad of death, rather than complicated by it.
We thank the reviewer for this note. The sentence has been altered to “ATC has been associated with high mortality, particularly when accompanied by hypothermia and acidosis; a difficult to treat state known as the ‘triad of death’.”
Line 48: “45 minutes to return”: do the authors have a reference for this? Or is it their anecdotal experience?
This is the experience of the authors at their respective hospitals, hence we have not supplied a reference. Nevertheless, we have added “anecdotally” to the sentence.
Line 49: “which remain unavailable in most centres”. Again, do the authors have a reference for this bold statement? I’m not sure that it is even true. And in which country? Do they mean internationally?
We appreciate this suggestion. We have reworded the phrase to “which are not available in all centres” which the authors have experienced to be the case.
Line 52: “blind formulaic management” sounds a bit pejorative. Do the authors mean empirical treatment?
Thank you for this comment. We have changed this phrase to “formulaic empirical management” to sound less pejorative. However, we want to emphasise that management of trauma patients is often based on formulae rather than individualised treatments guided by laboratory results.
Methods
Lines 80-81: “the population of patients with ATC and those requiring massive transfusion are substantially different.” This is not true, and not supported by reference 10.
We appreciate this comment. We have changed the sentence to “the population of patients with ATC and those requiring massive transfusion are not identical.” We believe this is supported by the reference.
In the inclusion criteria the authors say that “coagulopathy defined by International Normalised Ratio (INR), activated partial thromboplastin time (aPTT), thromboelastography (TEG), or rotational thromboelastometry (ROTEM) or fibrinogen levels.” But then later the authors include the Zane Perkins PhD thesis which uses PTr. The methods and results needs to match up better.
Thank you for this comment. Prothrombin time ratio (PTr) is the same as INR, thus we did not explicitly state it. It has now been added to the inclusion criteria section.
Results
I don’t understand how the authors have excluded some studies that were abstracts only (eligibility phase of the PRISMA diagram), but then in their final 6 studies they ended up with two abstracts (one had an associated poster and one had a PhD thesis). This doesn’t make sense. Did they contact all of the authors of the abstracts to see if there was an associated poster or PhD, or only those two? If so, why? Did the previous abstracts lack eligibility in other ways?
We appreciate this question and attempt to answer it here: the authors attempted to contact by email the authors of potentially relevant articles for which only abstracts were available. Indeed, had the full-text been available, they may have also met other exclusion criteria. At the point of screening however, the main obstacle was the lack of availability of a full-text, because of a lack of response or a lack of the existence of anything more than a conference abstract.
Does the second Tonglet study (2017) belong with the main studies of interest? It’s outcome is based on blood transfusion rather than coagulopathy. Therefore it is rightly included in this manuscript in terms of the validation of the earlier score, but is does not itself fulfil all the eligibility criteria. I would therefore say that the authors have identified 5 (not 6) studies that fulfilled all eligibility criteria.
The authors thank the reviewer for this insightful comment. We have altered the wording of the results section to reflect that the second Tonglet study was included for its similarity to the previous article. One additional article described the validation of a scoring system previously produced to predict ATC.
For the tables that reproduce the various scoring systems (Table 3 onwards), have the authors taken these directly from the studies of interest (in which case they should declare it), or have they adapted the tables, or made their own based on the individual studies? This needs to be more clear and avoid any accidental plagiarism
Thank you for this piece of advice. We have added to the Methods section the following sentence: “Tables describing the included scoring systems and tools have been reproduced from their original publications and cited as such.”
Line 192 “man” is a typo.
We thank the reviewer for noticing this error, it has been rectified.
Line 306: “The PACT score and the Bayesian model excluded these groups, which may produce a purer sample of patients with only “true” ATC”. I don’t understand what the authors mean here – they need to elaborate or delete this statement.
Thank you for this suggestion. We have altered the sentence: “…which may produce a purer sample of patients with only “true” ATC, as opposed to injured people with coagulopathy of other aetiology.”
Line 324 “INR >1.5 or aPTT >60sec is a comparatively high threshold for diagnosis of ATC”: do the authors have a reference to support this statement? Have they just taken the upper ends of the values quoted in the subsequent statements? This seems a bit arbitrary.
We believe this statement introduces the discussion of different definitions for ATC. The fact that different scoring systems use different definitions makes comparing the scores and the prevalence of ATC difficult.
Line 344 I don’t think the authors can declare that TEG and point of care tests are “ineffective and relatively uncommon” without some elaboration or citing some sources for this bold statement! There are many centres (for example in the USA) that use TEG and not ROTEM.
Thank you for this comment. We have deleted this phrase as it is indeed untrue.
Discussion
The authors attempt to discuss the various pros and cons for each of the scoring systems, but do not really propose any solutions. For example, could some of the scoring systems be combined? Or could a better scoring system be produced? Or should we chose one of these scoring systems and abandon the rest? Or do we need a different system for each category of patient? Is Bayesian technology the way forward? Are there some clinical variables common in the majority of these scoring systems, and if so, why? The authors only found a limited number of scoring systems and I am interested in the "so what" of this systematic search - this is the selling point to the reader, rather than just signposting to the scoring systems and summarising them individually.
The authors appreciate these ideas and suggestions. We have attempted to address these ideas further in the final paragraphs of the discussion.
Reviewer 2 Report
Thank you for the opportunity to review this interesting manuscript. It is a systematic review of early identification of acute traumatic coagulopathy using clinical prediction tools. The authors are to be congratulated on their findings and achievements on the exploration and analysis of the four prediction models to predict acute trauma coagulopathy.
In my opinion, the authors provided a sound scientific report. The objectives were clearly stated. The necessities of the analysis were adequately explained. The study method was adequately validated. The results clearly presented. The discussion pointed out the important findings. The conclusions appropriately based on the results and discussions.
I am appreciated for the fact and congratulated on authors’ achievement. The four clinical prediction tools the authors discussed in the study had their own strengths and limitations. The accuracy and generalization are needed further validation. I am appreciated if the authors may provide their proposal of potentially useful models, or discuss more how to improve and validate the current tools for clinical practice.
Author Response
Thank you for the opportunity to review this interesting manuscript. It is a systematic review of early identification of acute traumatic coagulopathy using clinical prediction tools. The authors are to be congratulated on their findings and achievements on the exploration and analysis of the four prediction models to predict acute trauma coagulopathy.
In my opinion, the authors provided a sound scientific report. The objectives were clearly stated. The necessities of the analysis were adequately explained. The study method was adequately validated. The results clearly presented. The discussion pointed out the important findings. The conclusions appropriately based on the results and discussions.
I am appreciated for the fact and congratulated on authors’ achievement. The four clinical prediction tools the authors discussed in the study had their own strengths and limitations. The accuracy and generalization are needed further validation. I am appreciated if the authors may provide their proposal of potentially useful models, or discuss more how to improve and validate the current tools for clinical practice.
The authors thank the reviewer for the kind words. We have attempted to address ideas for the future of this area in the final paragraph of the discussion.
Reviewer 3 Report
Acute traumatic coagulopathy (ATC) is a severe disorder for patients with major trauma. The authors attempted to find out some prehospital clinical prediction tool to predict ATC in this study. The authors find several prediction scores including COAST, TICCS, PACT and a Bayesian network model. Owing to the limited sample size, they only managed to perform the systematic review without a quantitative meta-analysis. The strength of this study is the detailed evaluation of each study, such as missing data management, appreciation of different populations.
Major weakness
The main weakness is that the authors only use the Newcastle-Ottawa Quality Assessment Form for Cohort Studies to evaluate the risk of bias tool. Although they claimed to use the TRIPOD (Transparent Reporting of a multivariable prediction model for Individual Prognosis Or Diagnosis) to evaluate each study, many potential biases are still not evaluated in the current format of this manuscript. For example, the authors did not comment on the different study design and definition of the gold standard (ATC) and their influence on the performance of these models. I recommend the authors to check the PROBAST (Prediction model Risk Of Bias ASsessment Tool) and the CHARMS (CHecklist for critical Appraisal and data extraction for systematic Reviews of prediction Modelling Studies) guidance to further evaluated they evaluation of the risk of bias. The performance evaluation of these models is incomplete. For example, only providing PPV or NPV in the TICCS study is not sufficient in evaluating a study, owing to the vulnerability of the parameters, which would be easily fluctuated in populations with different prevalences. The authors could either go back to look for other performance indicators or calculate them by hand using these numbers.Minor:
The authors claim to find a suitable model for prehospital use. However, they did not specify in their searching strategy, and the studies included are not devoted to prehospital use. The authors should revise their statement accordingly. Usually, the researchers would widen the search strategy when too few studies were found in a systematic review. In other words, using only the Mesh term to search would generate a too narrow searching strategy. Accordingly, I recommend that the authors shift from MeSH to text words to include more candidate studies. The text mining methods merit detailed description. The study design TICCS is not clear.Author Response
Acute traumatic coagulopathy (ATC) is a severe disorder for patients with major trauma. The authors attempted to find out some prehospital clinical prediction tool to predict ATC in this study. The authors find several prediction scores including COAST, TICCS, PACT and a Bayesian network model. Owing to the limited sample size, they only managed to perform the systematic review without a quantitative meta-analysis. The strength of this study is the detailed evaluation of each study, such as missing data management, appreciation of different populations.
Major weaknesses
The main weakness is that the authors only use the Newcastle-Ottawa Quality Assessment Form for Cohort Studies to evaluate the risk of bias tool. Although they claimed to use the TRIPOD (Transparent Reporting of a multivariable prediction model for Individual Prognosis Or Diagnosis) to evaluate each study, many potential biases are still not evaluated in the current format of this manuscript. For example, the authors did not comment on the different study design and definition of the gold standard (ATC) and their influence on the performance of these models. I recommend the authors to check the PROBAST (Prediction model Risk Of Bias ASsessment Tool) and the CHARMS (CHecklist for critical Appraisal and data extraction for systematic Reviews of prediction Modelling Studies) guidance to further evaluate the risk of bias.
The authors are grateful to the reviewer for suggesting this. We have used the PROBAST tool instead of the Newcastle-Ottawa tool and supplied results in Table 1.
The performance evaluation of these models is incomplete. For example, only providing PPV or NPV in the TICCS study is not sufficient in evaluating a study, owing to the vulnerability of the parameters, which would be easily fluctuated in populations with different prevalences. The authors could either go back to look for other performance indicators or calculate them by hand using these numbers.
The authors appreciate this comment and suggestion. We have calculated and included the sensitivity and specificity of the TICCS validation score.
Minor weaknesses
The authors claim to find a suitable model for prehospital use. However, they did not specify in their searching strategy, and the studies included are not devoted to prehospital use. The authors should revise their statement accordingly.
We thank the reviewer for noticing this oversight: we changed our intention from finding prehospital tools to early prediction tools, but failed to change one sentence in the abstract. This has now been rectified.
Usually, the researchers would widen the search strategy when too few studies were found in a systematic review. In other words, using only the Mesh term to search would generate a too narrow searching strategy. Accordingly, I recommend that the authors shift from MeSH to text words to include more candidate studies.
We appreciate this suggestion. Our search strategy included both MeSH terms and text words because we had concerns that our search would indeed be too narrow. Our search strategy, shown in Appendix A, demonstrates our broad search. This is also corroborated by the number of results we found with this search strategy.
The text mining methods merit detailed description.
Thank you for this suggestion. We have described in more detail: “two authors read the reference lists of included articles and screened any potentially relevant articles which had not yet been found for inclusion.”
The study design TICCS is not clear.
We understand this comment from the reviewer, but we have been unable to come up with any way of making the study design clearer. We hope this will suffice, and readers can refer to the cited article for more information.
Round 2
Reviewer 1 Report
All my comments have been addressed and the manuscript is much improved. Only one small comment - the authors should not use the word 'manuscript' in the paper, since once it is published it is no longer a manuscript... the word 'study' or 'paper' should be used instead (eg page 4 line 117)
Author Response
All my comments have been addressed and the manuscript is much improved. Only one small comment - the authors should not use the word 'manuscript' in the paper, since once it is published it is no longer a manuscript... the word 'study' or 'paper' should be used instead (eg page 4 line 117)
Thank you for this piece of information. All erroneus uses of the word 'manuscript' have been replaced.
Reviewer 3 Report
The text mining methods merit detailed description.
Thank you for this suggestion. We have described in more detail: “two authors read the reference lists of included articles and screened any potentially relevant articles which had not yet been found for inclusion.”
Further question: the current description does not fit the term "text mining", which should be removed from the manuscript.
Author Response
The text mining methods merit detailed description.
Thank you for this suggestion. We have described in more detail: “two authors read the reference lists of included articles and screened any potentially relevant articles which had not yet been found for inclusion.”
Further question: the current description does not fit the term "text mining", which should be removed from the manuscript.
Thank you for your comments. This term has been removed from the text.